# Adherence to voluntary UK sugar, salt, and calorie reduction targets in the highest-grossing restaurant chains: A cross-sectional study

Alice O'Hagan *, Rachel Pechey, Hannah Forde, Lauren Bandy

Nuffield Department of Primary Care Health Sciences, University of Oxford, Oxford, United Kingdom

* alice.ohagan@phc.ox.ac.uk

## Abstract

### Background

To address high rates of diet-related disease, the UK Government has a series of voluntary targets for retailers, manufacturers, and the out-of-home sector (e.g., restaurants), to reduce the sugar, salt, and calorie content of food products. The sugar targets were intended to be met in 2020, the salt targets in 2024, and the calorie targets in 2025 (extended from 2024 due to Covid-19). There is limited evidence for how the out-of-home sector is performing against these targets, and individual company responses have not been evaluated. This study aimed to assess adherence to UK Government's sugar, salt, and calorie reduction targets for menu items offered by the 21 highest-grossing restaurant chains in 2024.

### Methods and findings

Nutritional information was collected from restaurants' online menus. Mean/median sugar, salt, and calorie content, per 100 g and per serving, was calculated for each restaurant and food subcategory. Sugar, salt, and calorie content for each menu item was compared against the UK Government's targets, and the proportion of menu items meeting (i) each and (ii) every applicable target, was calculated for each restaurant and food subcategory.

Three thousand ninety-nine menu items were included. Across all restaurants, 61% of menu items met their calorie targets, 58% met their salt targets, 36% met their sugar targets, and 43% met all of their applicable targets. Six of the 12 food sub-categories, and nine of the 21 restaurants, had over 50% of menu items meeting all of their applicable targets. Menu items from Papa John's were the lowest adhering for the calorie (35%) and salt (8%) targets, and menu items from Burger King, KFC, Nando's, and Vintage Inns were the lowest adhering for the sugar targets (0%). Menu items from pizza restaurants had the lowest adherence to all applicable targets (32%

**Data availability statement:** The dataset used to conduct the analyses in this article is publicly available, deposited in the Oxford University Research Archive under CC-BY license (DOI: https://doi.org/10.5287/ora-6raddkrg9).

**Funding:** AOH and RP are supported by the NIHR Oxford Health Biomedical Research Centre (https://oxfordhealthbrc.nihr.ac.uk/). RP is also supported by the Royal Society and Wellcome Trust (Sir Henry Dale fellowship; 222566/Z/21/Z; https://www.royalsociety.org/grants/henry-dale/). HF is funded by the SHIFT: Sustainable and Healthy Interventions for Food Transitions project, which is funded by the Wellcome Trust (grant reference 227132/Z/23/Z; https://wellcome.org/research-funding/funding-portfolio/funded-grants/shift-sustainable-and-healthy-interventions-food), and by the COPPER project, which is funded by the National Institute for Health and Care Research (NIHR) Public Health Research programme (grant reference NIHR133887; https://fundingawards.nihr.ac.uk/award/NIHR133887). LB is supported by the NIHR Applied Research Collaboration (ARC) Oxford and Thames Valley (https://www.arc-ox-tv.nihr.ac.uk/). The funders had no role in study design, data collection and analysis, decision to publish, or preparation of the manuscript.

**Competing interests:** The authors have declared that no competing interests exist.

**Abbreviations:** NCD, non-communicable disease; OOH, out of home;STROBE, Strengthening the Reporting of Observational Studies in Epidemiology.

overall) out of all the restaurant types, but items offered by restaurants with similar menu foci were also found to vary in their adherence.

We were unable to account for heterogeneity in item-level sales due to the lack of accessible sales data from the out-of-home food sector, and therefore we could only assess performance against the targets for available items as opposed to purchased items.

## Conclusions

Our findings suggest that while menu items from certain restaurant types appear to perform worse than others against the sugar, salt, and calorie targets, items from restaurants with similar menu portfolios also vary in their adherence, highlighting the potential for restaurants to improve the nutritional quality of their products without changing their menu focus. Our study demonstrates that there is low adherence to voluntary schemes across the out-of-home sector, and therefore mandatory regulations may be a more effective approach to improving the nutritional quality of out-of-home food.

## Introduction

The purchasing and consumption of foods high in energy, saturated fat, free sugars, and salt, is associated with an increased risk of obesity and diet-related non-communicable diseases (NCDs) [1]. Globally, diet-related NCD prevalence is high, with approximately 40% of cardiovascular disease mortality between 1990 and 2019 attributable to dietary risk factors [2]. The same is true for the UK specifically, with poor diet being a leading cause of death and ill health [3].

Food reformulation initiatives are a common approach to NCD prevention across the globe; 68 countries have a salt reduction policy in place [4], with countries such as the UK, Brazil, New Zealand, and USA, having further initiatives for sugar reduction [5], with the majority of such schemes being voluntary [4,5]. Modelling studies highlight the potential for such policies to significantly reduce the incidence of diet-related diseases, such as obesity and cardiovascular disease [6,7]. The UK Government first introduced their salt reduction programme in 2004, following advice from the Scientific Advisory Committee on Nutrition to reduce the population average salt intake by approximately 10%, to 6 g a day. The first reduction and reformulation targets for industry were published in 2006 and have since been revised on several occasions, with the most recent revision in 2020 (for targets to be met by 2024) covering 84 category-specific salt targets for grocery foods (e.g., Ready Meals), and 24 for the out-of-home (OOH) sector (e.g., Seasoned Fries) [8]. Evidence suggests that the salt reduction programme has been successful, with average urinary sodium reducing by approximately 2% each year from 2004 to 2011, and significant reductions being observed in the salt content of food products sold by UK retailers [9].

Complementary to the salt reduction programme, the UK Government published targets for sugar and calories in 2017 and 2020, respectively [10,11]. The sugar reduction programme aimed to achieve a 20% reduction in the sugar content of 14 food categories that contribute most to sugar intake in children, across all sectors of the food industry, by 2020 [10]. The calorie reduction programme aimed to achieve a 10% reduction by retailers and manufacturers across nine food categories, and a 20% reduction by the OOH sector across seven food categories (with four common categories), in the calorie content of products sold, by 2025 (extended from 2024 due to the Covid-19 pandemic) [11]. Governmental progress reports indicate that within the retailer and manufacturer sector, there has been a 3.5% reduction in the sugar content of products sold between 2015 and 2020 [12], and category-level decreases of up to 2.4% in the calorie content of products sold between 2017 and 2021 [13]. Their findings suggest comparatively less progress has been made in the OOH sector, with only a 0.2% reduction in the sugar content of products sold between 2017 and 2020, and category-level increases of up to 2.3% in the calorie content of products sold between 2017 and 2021 [12,13]. However, since the most recent data included in the sugar, salt, and calorie reduction progress reports was from 2020, 2018, and 2021 respectively, these conclusions may have since changed.

A considerable proportion of UK individuals' weekly food intake comes from takeaway or restaurant meals [14]. The UK Diet and Nutrition Survey finds 72% of respondents reported purchasing food or drink from an OOH establishment in the past week, with the largest proportion of these consumers being children aged 11–18 years old [15]. Eating OOH is becoming more popular, with a 159% increase in per-person expenditure on eating out from 2021 to 2022 in the UK [16], and an additional 3.5% increase to 2023 [17], potentially due to its perceived lower cost and higher convenience than preparing food at home [18]. The OOH sector is dominated by a small number of multinational companies who operate chained food outlets (hereafter 'restaurants'; see Table 1). In 2023, sales from chained food service outlets reached £35.1 billion in the UK, increasing by nearly £8 billion over the previous two years [19], showcasing the significant influence these companies have over diet and health.

The UK Government's sugar, salt, and calorie reduction targets are voluntary, leaving the responsibility for meeting the targets to individual companies, yet there is limited evidence for how food companies have responded to the reduction programmes. One study looking at company-level adherence to the sugar reduction targets among UK manufacturers found that of the top 10 companies across 5 target food categories, just under half met the interim 5% sugar reduction targets for 2018 [20]. Similar work conducted in the OOH sector found that only four out of 48 brands reduced the sugar content of their desserts by at least 20% from 2018 to 2020, and only half of which significantly reduced their calorie content as well [21].

There is a need for a more comprehensive assessment of company-level adherence to the targets. Furthermore, fewer studies in the broader literature have assessed the nutritional quality of foods in the OOH sector compared to the retailer and manufacturer sector, due to OOH nutrition data being less easily accessible. Evaluating industry's response to the reduction programmes will improve transparency around companies' commitment to population health, help with enforcement of comparable schemes in the future, and underpin other approaches to incentivising healthier food provision (e.g., investment decision-making based on food healthiness). The introduction of mandatory reporting of healthy sales from large companies, as pledged in the NHS 10 Year Plan [22], will contribute to improving transparency and potentially incite greater commitment from companies to meet nutritional guidelines. This study aimed to assess the nutritional content and adherence to the UK Government's sugar, salt, and calorie reduction targets, for food items offered by the highest-grossing restaurant chains in the UK, in 2024.

## Methods

We assessed the nutritional content of food items offered in 2024 by the 21 highest-grossing restaurant chains in the UK. We collected nutrition information for restaurants' menu items directly from their websites. Each menu item was categorised by the sugar, salt and calorie reduction target groups and their nutrient values compared to the

**Table 1. A summary of characteristics for each restaurant, including their affiliated company, type, 2022 sales value, number of menu items included, and number of subcategories included. Restaurants presented in descending order by 2022 Sales Value.**

| UK Brand Name | Company Name | Restaurant Type | 2022 Sales Value (GBP Million) | Number of menu items included | Number of subcategories included (maximum 12) |
|---|---|---|---|---|---|
| McDonald's (McDonald's Corp) | Various franchisees | Burger | 3086.40 | 111 | 10 |
| KFC (Yum! Brands Inc) | Various franchisees | Chicken | 1396.90 | 76 | 10 |
| Domino's Pizza (Domino's Pizza Inc) | Domino's Pizza Group Ltd | Pizza | 1347.00 | 241 | 6 |
| Greggs (Greggs Plc) | Greggs Plc | Sandwich | 1204.00 | 174 | 10 |
| Costa Coffee (Coca-Cola Co, The) | Coca-Cola Co, The | Sandwich | 1045.00 | 123 | 7 |
| Pret a Manger (Pret A Manger (Europe) Ltd) | Pret A Manger (Europe) Ltd | Sandwich | 1006.30 | 151 | 8 |
| Nando's (Nando's Group Holdings Ltd) | Nando's Chickenland Ltd | Chicken | 831.20 | 90 | 10 |
| Subway (Doctor's Associates Inc) | Various franchisees | Sandwich | 599.50 | 80 | 7 |
| Burger King (Restaurant Brands International Inc) | Various franchisees | Burger | 562.10 | 40 | 5 |
| Starbucks (Starbucks Corp) | Starbucks Coffee Holdings (UK) Ltd | Sandwich | 544.80 | 49 | 5 |
| Pizza Hut (Yum! Brands Inc) | Pizza Hut UK Ltd | Pizza | 388.40 | 330 | 6 |
| Papa John's (Papa John's International Inc) | Papa John's International Inc | Pizza | 356.60 | 180 | 6 |
| Caffé Nero (Italian Coffee Holdings Ltd, The) | Italian Coffee Holdings Ltd, The | Sandwich | 341.20 | 108 | 8 |
| Wagamama (Restaurant Group Plc, The) | Restaurant Group Plc, The | Other Main | 339.20 | 105 | 6 |
| PizzaExpress (PizzaExpress (Restaurants) Ltd) | PizzaExpress (Restaurants) Ltd | Pizza | 291.60 | 247 | 8 |
| Harvester (Mitchells & Butlers Plc) | Mitchells & Butlers Plc | Other Main | 208.30 | 175 | 9 |
| Hungry Horse (Greene King Plc) | Greene King Plc | Other Main | 181.30 | 247 | 10 |
| Prezzo (Prezzo Plc) | Prezzo Plc | Other Main | 164.20 | 218 | 9 |
| Toby Carvery (Mitchells & Butlers Plc) | Mitchells & Butlers Plc | Other Main | 130.50 | 214 | 8 |
| Vintage Inns (Mitchells & Butlers Plc) | Mitchells & Butlers Plc | Other Main | 123.70 | 73 | 8 |
| Leon (Leon Restaurants Ltd) | Leon Restaurants Ltd | Other Main | 80.70 | 67 | 11 |

relevant targets. The average kcal, salt, sugar per 100 g and per serving, was also calculated for each restaurant and food subcategory. The study protocol was published on Open Science Framework prior to data analysis [23] (S1 File). This study is reported as per the Strengthening the Reporting of Observational Studies in Epidemiology (STROBE) guideline (S1 Table—STROBE checklist). Ethical approval was not required for this study as it does not involve human participants.

## Identifying restaurants

This study focuses on the highest-grossing restaurant chains in the UK, as these restaurants have the largest impact on diets within the OOH food sector. We identified the 21 highest-grossing chained restaurants in the UK using the latest (2022) sales data from Euromonitor International, a privately-owned market research company. Euromonitor's data portal Passport GMID was accessed via the Bodleian Library, University of Oxford [24]. We selected restaurants that Euromonitor defined as "Consumer Foodservice", including "cafés/bars, full-service restaurants, limited-service restaurants,

self-service cafeterias and street stalls/kiosks". "Full-service" refers to sit-down restaurants with table service, while "limited-service" refers to fast food or take-away outlets.

We excluded restaurants that had separate menus for each individual location to avoid introducing considerable heterogeneity to nutritional information at the restaurant level. Restaurants that only provided nutrition information per ingredient (e.g., bun, patty, cheese) rather than per item (e.g., complete burger) were also excluded, as it was unclear what a full menu item would consist of.

We initially aimed for a sample of 20 restaurants. Subway was excluded at the start of the data collection phase as it provided nutrition information for individual ingredients only. However, new menu-item nutrition information was subsequently published after our data collection had started, and therefore Subway was reinstated and a sample of 21 restaurants' data was collected.

## Data collection

Menu data was collected directly from restaurants' UK websites in February and March of 2024 (May 2024 for Subway). It was collected primarily through PDF menu documents or menu webpages. Where a restaurant provided multiple PDFs, data was only collected from those labelled as containing 'core' (or equivalent) menu items. Limited-time offer menu items (e.g., seasonal) were only included in data collection if they were present in the 'core' menus. For pizza restaurants that offered multiple crust options, only nutrition information from the default option (e.g., classic crust) was collected, but all size variations (e.g., small, medium, large) were included. Meals that combined several food items that could be purchased separately, were also included as their own menu item.

Data in PDFs were extracted either by copying and pasting the data into Excel or by using an online data extraction software (Smallpdf [25]). Where data had to be extracted from menu webpages, we used the web scraping tool Octoparse (v8 desktop [26]) to extract the menu data into an Excel sheet. The extracted data was crosschecked against the source material by AOH. The data collection approach for each restaurant is detailed in S2 Table.

The data extracted included restaurant name, product name, nutritional information, and serving size, where available. Nutritional information was collected per 100 g and per serving wherever given, and included kcal, kJ, fat, saturated fat, carbohydrates, sugar, protein, fibre, salt, and sodium.

## Categorising menu items

All menu items were categorised twice. First, author AOH assigned all menu items to one of the following 12 common subcategories: Pizzas, Burgers, Chicken, Other Mains, Children's Meals, Salads, Sandwiches, Potato Sides, Other Sides, Breakfast Items, Desserts, and Sauces. This provided a consistent framework in which we could compare the nutritional content of similar menu items across restaurants. Hereafter, this categorisation will be referred to as an item's 'subcategory'. Categorisation criteria for each subcategory is detailed in S3 Table.

Second, menu items were also matched to the OOH food categories as defined by the technical guidance for the sugar, salt, and calorie reduction targets [8,10,11]. The target-specific categories are not inclusive of all menu items and the guidance states that there should be no overlap between the calorie and sugar targets. Therefore, a menu item could be (i) ineligible for all three target types, (ii) eligible for only one of the three target types, (iii) eligible for a salt and calorie target but not a sugar target, or (iv) eligible for a salt and sugar target but not a calorie target. To align with the guidance that an item should not have both a calorie and a sugar target, we first categorised items by the sugar targets, and then categorised the remaining items by the calorie targets, as the sugar reduction target programme preceded the calorie target programme. The salt reduction programme provided separate targets for the OOH sector and the retailer and manufacturer sector. In line with the technical guidance for the salt targets, menu items were first categorised against the OOH targets, and any remaining uncategorised items were categorised against the retailer and manufacturer targets.

A random 10% of menu items' categorisations were checked by two co-authors (LB and HF) who had not been involved in the original categorisation process. A small subset of menu items was re-categorised and changes were applied to the complete dataset where relevant, followed by a final round of checking by author AOH.

We divided our 21 restaurants into the following five 'restaurant types', based on the predominant main meal subcategory that the restaurant offered: Burger restaurants, Chicken restaurants, Pizza restaurants, Sandwich restaurants, and Other Mains restaurants. The 12 subcategories were also divided into 'Mains' and 'Sides/Extras', with 'Desserts' in its own category.

## Analysis

Data analyses were conducted in R (version 4.4.2) [27], using the following packages: readxl [28], dplyr [29], tidyr [30], stringr [31], tidyverse [32], zoo [33], writexl [34], reshape [35], ggplot2 [36].

### Missing data

We excluded menu items where all three of the nutrients of interest (kcal, salt, and sugar), were missing. For menu items that were missing two or fewer of the nutrients of interest, where possible, we calculated the missing values using the following formulas:

$$kcal = kJ/4.184 \ (and \ vice \ versa)$$

$$kcal = (Protein\ (g) * 4) + (Fat\ (g) * 9) + (Carbohydrate * 3.75)$$

$$Sodium\ (mg) = (Salt\ (g) * 1000)/2.5 \ (and \ vice \ versa)$$

In order to assess an item's adherence to the Government's reduction targets, nutrition information was needed per 100 g for the sugar targets, per serving for the calorie targets, and in both formats for the salt targets. If per 100 g, per serving, or serving size information was missing, the missing value was calculated from the provided information using the following formulas:

$$Per\ 100\ g = Per\ serving/(Serving\ Size/100)$$

$$Per\ Serving = (Serving\ size/100) * Per\ 100\ g$$

$$Serving\ Size = (100/Per\ 100g) * Per\ Serving$$

Here, 'Serving Size' refers to the weight of the menu item in grams, and 'Per 100 g' and 'Per Serving' refer to nutrient content (e.g., salt content in 100 g or in a single serving of a menu item).

Where it was not possible to calculate serving size (only per 100 g or only per serving nutrient information was provided), the mean serving size for the menu item's subcategory was used instead. For example, if the serving size for a burger was not provided, the mean serving size of menu items within the 'Burger' subcategory was used instead.

### Average nutrient content

The mean and median kcal, sugar, salt, and fat content was calculated for each restaurant and subcategory. Averages were calculated using nutrient information provided per 100 g, per reported serving size, and per subcategory average

serving size. 'Reported serving size' refers to the serving size of a menu item as reported by the restaurant, whereas 'subcategory average serving size' refers to the average serving size of menu items within a subcategory (e.g., the average serving size for menu items within the 'Burger' subcategory).

For Pizzas, average nutrient content per serving was calculated using the serving sizes provided by individual restaurants. Papa John's provided per serving information per slice, Pizza Hut and Domino's provided per serving information per person sharing (e.g., medium pizza is shared between two people, so one pizza would count as two servings), and the remaining restaurants provided per serving information per whole pizza.

### Adherence to targets

The sugar, salt, and calorie content of each product was compared to their matched category's target value. S4 Table shows the range of target values that menu items had to meet, within each subcategory. Where a menu item's sugar, salt, or calorie content was equal to or lower than the target value, they were deemed to have met the target. The proportion of each restaurant and subcategory's menu items that met the targets was expressed as a percentage of their total number of menu items eligible for the respective target.

### Sensitivity analyses

We repeated the primary analyses for mean nutrient content and target adherence, but with limited-time offer menu items that featured on the main menu, excluded.

We repeated the primary analyses, but for items where the subcategory average serving size was used to impute nutritional content (as the restaurant did not provide it), we instead used the lower quartile and upper quartile subcategory serving sizes to impute nutritional content, to check the robustness of our findings to altering our imputation strategy.

### Deviations from protocol

We had planned to use ANOVAs to determine whether the average nutrient content differed significantly across restaurants and subcategories, and logistic regression models to determine whether a menu item's restaurant or subcategory was a predictor of their adherence to the sugar, salt, and calorie reduction targets. We have opted instead for a descriptive presentation of the results, as the dataset is not a random sample from a wider population of menu items. This means that any differences in nutrient content or target adherence that we observed between restaurants or subcategories, are in fact real, and not attributable to sampling variation. As a result, inferential statistical testing is not warranted in this case.

As a sensitivity analysis, we planned to repeat the primary analysis for target adherence but using 'as sold' nutrition information instead of 'per serving' information. For example, if a restaurant reports that a menu item contains two servings, our primary analysis would have assessed target adherence based on a single serving (being 'per serving'), while our sensitivity analysis would assess it as the whole menu item ('as sold'). The aim of this analysis was to remove the influence of individual restaurants' reporting of serving size, to allow for a consistent assessment of nutritional quality across restaurants. On examination of the data, we found for Pizzas in particular, the suggested serving sizes and reporting of per serving nutrition information lacked consistency across restaurants. For example, Papa John's provided their nutrition information per slice without explicitly stating how many slices equate to a single serving, while Pizza Express provided information per whole pizza with no suggestions for how many servings it contained. The technical guidance for the salt and calorie targets attempts to account for this, with the salt targets being applied per slice or per whole pizza depending on the style of pizza (takeaway or Italian-style), and the calorie targets being applied per serving for 'sharing' pizzas (defined as large and above, or 11.5″ and above). Our analysis followed this guidance, with assumptions for serving size being made to apply the calorie targets where information was not provided. For example, Papa John's large and extra-large pizzas were assumed to contain three and four servings respectively, as they did not provide their own

suggestions, and Italian-style pizzas were assumed to be a single serving. For analyses with the outcome of average nutrient content per serving, serving sizes for Pizzas were as provided by the restaurants.

We planned to conduct a sensitivity analysis in which we would repeat all primary analyses, but for menu items that did not have a serving size reported, we would use an applicable serving size from the Food Standard's Agency 'Food Portion Sizes' handbook [37]. Upon further investigation, we concluded that the suggested serving sizes from this handbook would not be applicable to OOH menu items, as they were often provided per meal component rather than complete meal (e.g., suggested serving size provided for a burger patty and a bun, rather than a whole burger), and therefore this analysis was not conducted.

The serving size imputation sensitivity analysis was not pre-registered, but was included to test the robustness of our approach to dealing with missing data.

The planned exploratory analysis comparing the same menu items reported in 2022 with 2024 was not conducted due to the small sample size of products present on menus in both years.

As the targets are food only, and the same top-selling brands of drinks appear on the majority of restaurant's menus, drinks were excluded.

The planned analysis using the UK Ofcom/FSA Nutrient Profile Model as an additional assessment of nutritional quality, will be presented in a separate paper.

## Results

A total of 3,099 menu items across 21 restaurants were included in this study. Originally, 5,435 menu items were collected. Of the 2,336 items excluded, 453 were duplicates, 103 did not have the required nutritional information, 825 were pizzas with non-default crust options, and 955 were drinks. One menu item was missing a sugar value, so this item was excluded from analyses where sugar content is the outcome. For 1,630 out of 3,099 menu items, no serving size was provided and only one format of nutrient content was provided (either per 100 g or per serving), and therefore the subcategory average serving size was used in calculating the missing nutrient content. S5 and S6 Tables provide the number of affected products, and the mean, median, and lower and upper quartiles for serving size (g) (from the 1,469 where it was provided/calculated), for each restaurant and subcategory.

The mean number of menu items per restaurant was 148, although this varied widely from 40 items for Burger King to 330 items for Pizza Hut (Table 1). The number of subcategories offered by each restaurant ranged from 5 for Starbucks and Burger King, to 11 for Leon. Some restaurants had large menus that focussed on a smaller number of subcategories, such as Pizza Hut and Domino's, while others had smaller menus with more diverse types of items, such as Leon (Fig 1).

### Average nutrient content

**By subcategory.** Per 100 g, mean nutrient content across all menu items was 277 kcal, 1.1 g salt, and 9.5 g sugar. Per serving, mean nutrient content across all menu items was 450 kcal, 2.0 g salt, and 10.9 g sugar.

Desserts had the highest mean calorie (409 kcal) and sugar (34.2 g) content per 100 g across all subcategories, while Sauces had the highest salt content per 100 g (2.2 g) (Fig 2). Per serving, Other Mains had the highest calorie content (756 kcal) and joint highest salt content with Pizzas (3.4 g), and Desserts had the highest sugar content (26.2 g) (Fig 3). Means, Medians, and Standard Deviations for nutrient content per 100 g, per reported serving, and per subcategory average serving, for each subcategory, are found in S7–S10 Tables.

**By restaurant.** Menu items from Caffé Nero had the highest mean calorie content per 100 g (368 kcal) across all restaurants, while menu items from Prezzo had the highest salt content (1.8 g), and menu items from Costa had the highest sugar content (22.0 g) (Fig 4). Per serving, menu items from Prezzo had the highest calorie (644 kcal) and salt content (3.7 g) and menu items from Harvester had the highest sugar content (17.2 g) (Fig 5). Means, Medians, and

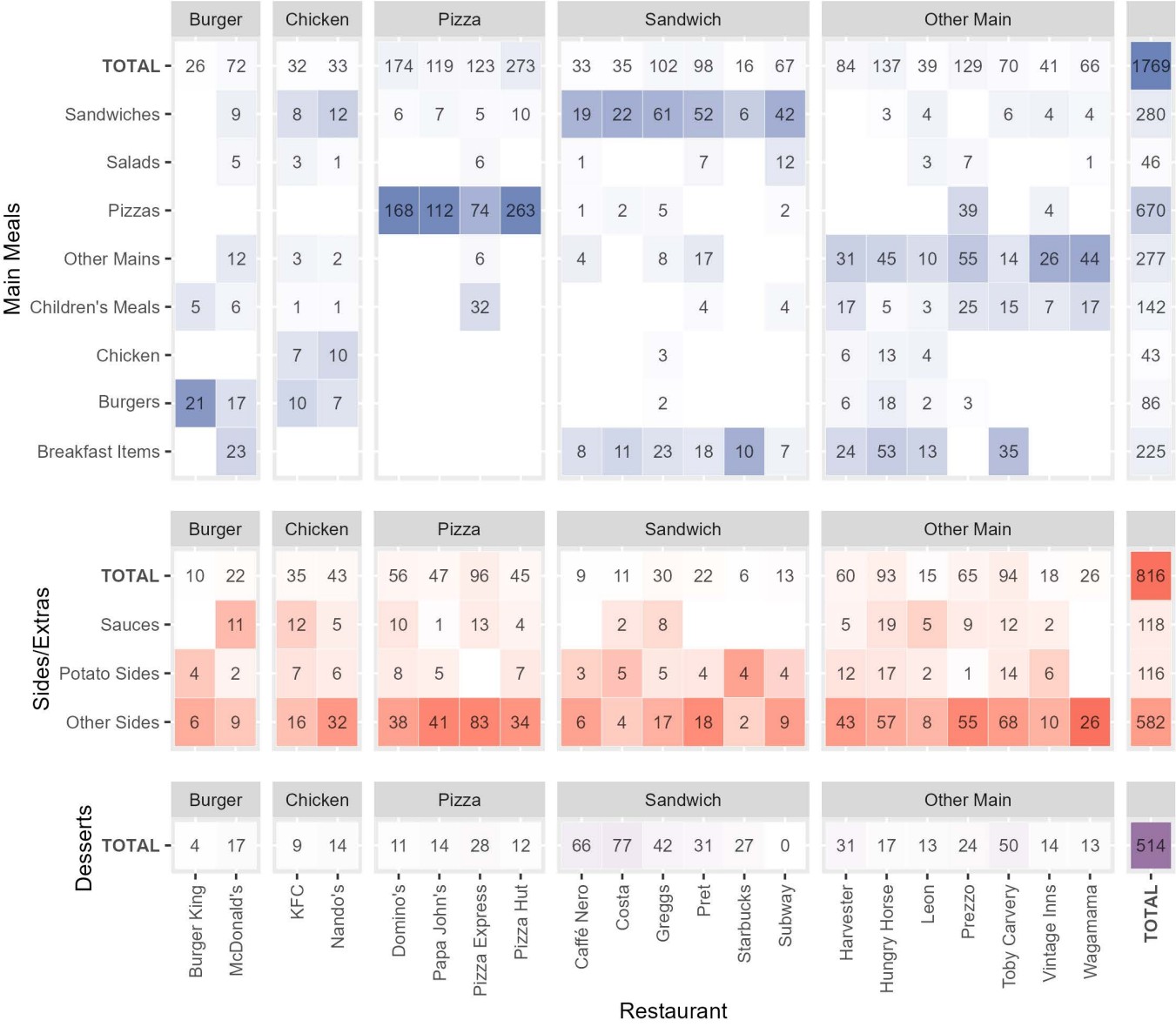

**Fig 1. The number and proportion of menu items by restaurant and subcategory.** The values in boxes show the number of menu items belonging to that restaurant-subcategory. Main meals are coloured in blue, sides/extras are coloured in red, and desserts are coloured in purple. The darkness of each colour indicates the proportion of menu items from each restaurant that belong to that subcategory (calculated separately for mains, sides/extras, and desserts, with the 'total' rows calculated across mains, sides/extras, and desserts), where a darker colour indicates a higher proportion. KFC and Nando's were characterised as 'Chicken' restaurant type despite majority of main meal menu items belonging to 'Sandwich' or 'Burger' subcategories, as these menu items were majority chicken burgers and chicken wraps. The same principle was applied to restaurants with the majority of their main menu items being 'Breakfast Items'.

Standard Deviations for nutrient content per 100 g, per reported serving, and per subcategory average serving, for each restaurant, are found in S11–S14 Tables.

Averaging across menu items within restaurant groups, menu items from the Sandwich group had the highest mean calorie (293 kcal) and sugar content per 100 g (13.4 g), and menu items from the Pizza group had the highest mean salt

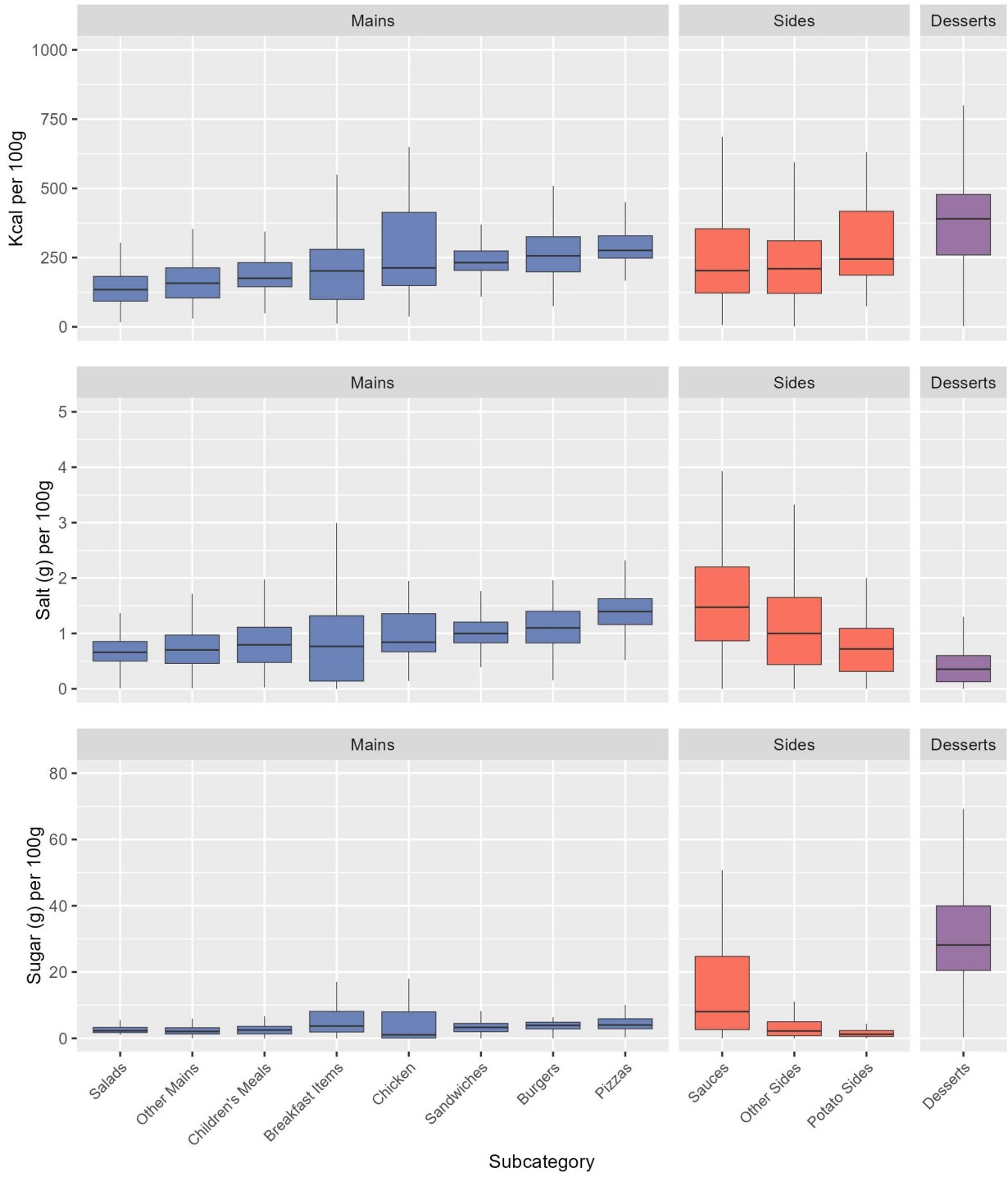

**Fig 2. The distribution of kcal, salt, and sugar content per 100 g for menu items by subcategory.** Subcategories are ordered ascendingly by median kcal per 100 g. Mid-lines represent the median, the box represents the inter-quartile range, and whiskers represent the range. Main meal subcategories are coloured in blue, Side categories are coloured in red, and Desserts are coloured in purple.

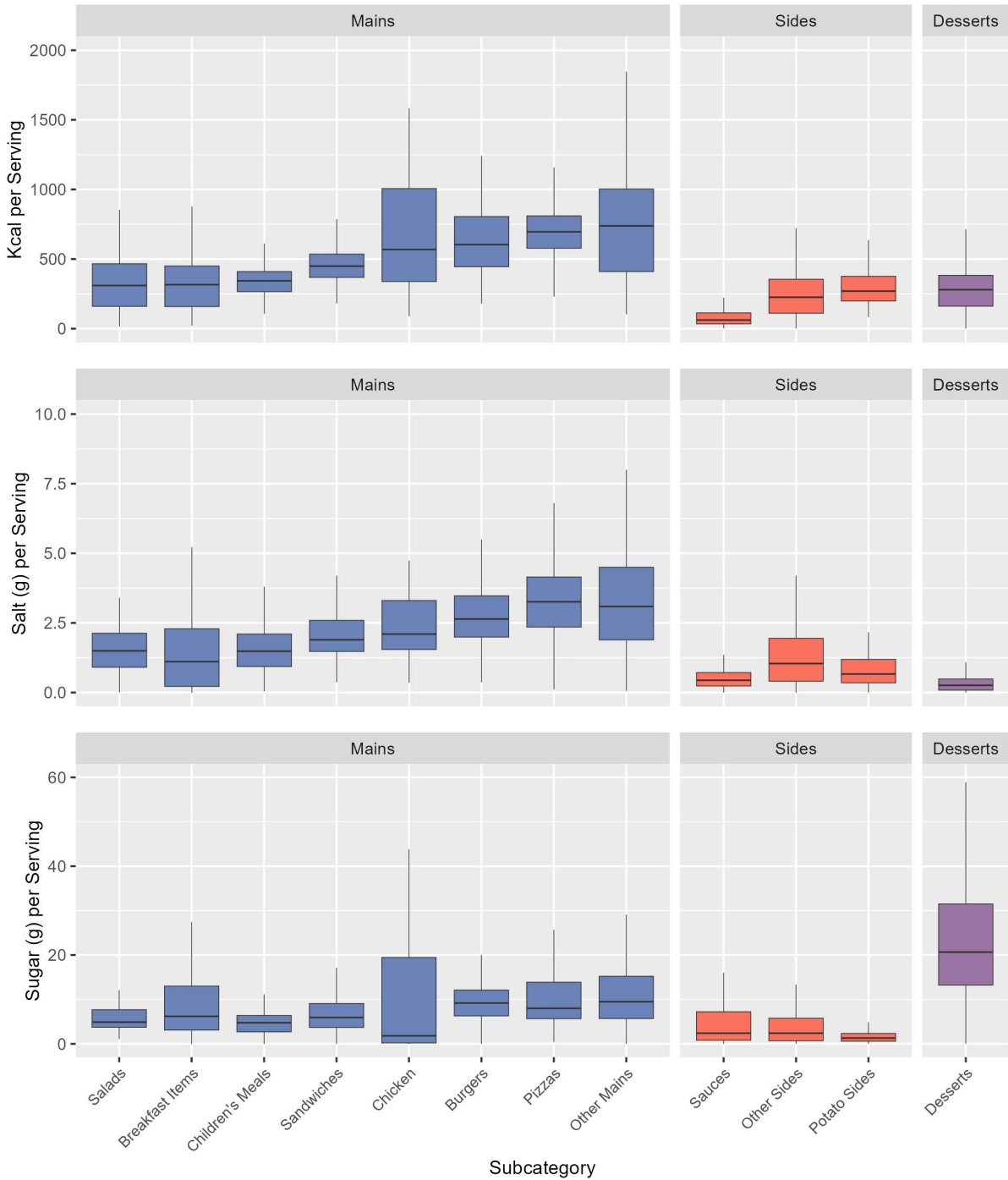

**Fig 3. The distribution of kcal, salt, and sugar content per serving for menu items by subcategory.** Subcategories are ordered ascendingly by median kcal per serving. Mid-lines represent the median, the box represents the inter-quartile range, and whiskers represent the range. Main meal sub-categories are coloured in blue, Side categories are coloured in red, and Desserts are coloured in purple.

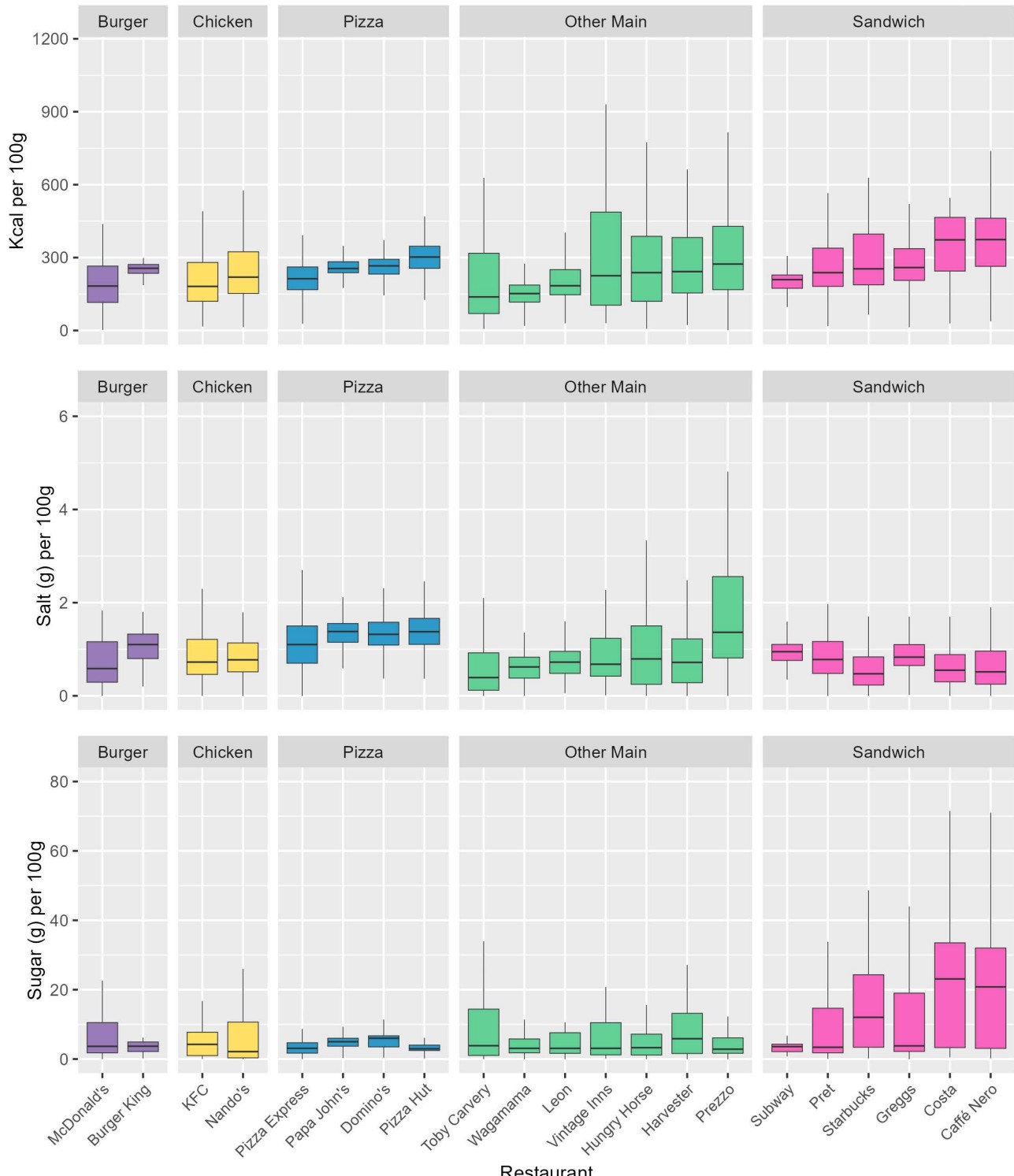

**Fig 4. The distribution of kcal, salt, and sugar content per 100 g for menu items by restaurant.** Restaurants are ordered ascendingly by median kcal per 100 g. Mid-lines represent the median, the box represents the inter-quartile range, and whiskers represent the range. Burger restaurants are coloured in purple, Chicken restaurants in yellow, Pizza restaurants in Blue, Other Main restaurants in green, and Sandwich restaurants in pink.

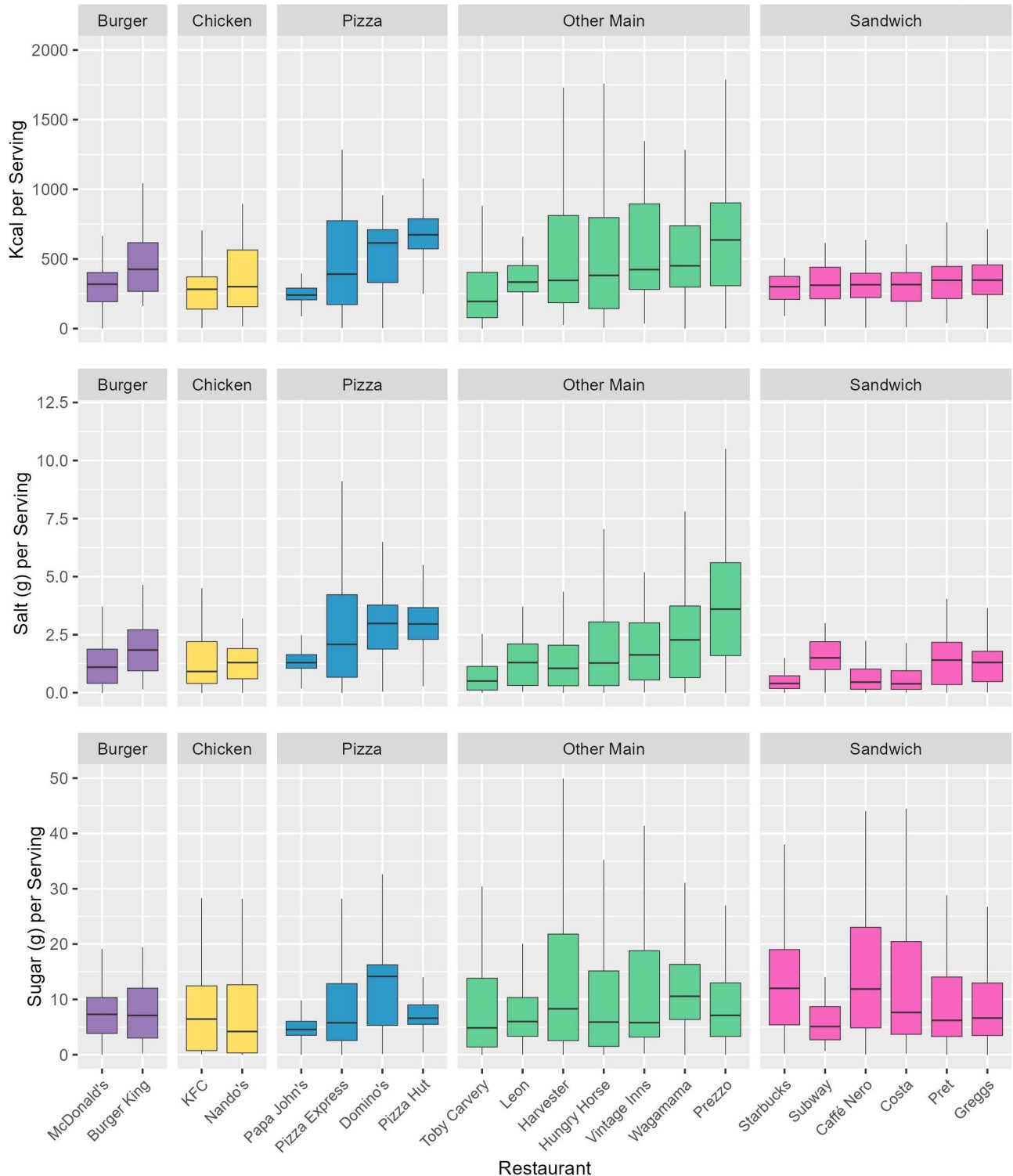

**Fig 5. The distribution of kcal, salt, and sugar content per serving for menu items by restaurant.** Restaurants are ordered ascendingly by median kcal per serving. Mid-lines represent the median, the box represents the inter-quartile range, and whiskers represent the range. Burger restaurants are coloured in purple, Chicken restaurants in yellow, Pizza restaurants in Blue, Other Main restaurants in green, and Sandwich restaurants in pink.

content (1.4 g). Menu items from the Pizza group had the highest mean calorie (515 kcal) and salt content per serving (2.6 g), and menu items from the Other Mains group had the highest mean sugar content (12.6 g). Means, Medians, and Standard Deviations for nutrient content per 100 g, per reported serving, and per subcategory average serving, for each restaurant group, are found in S15–S18 Tables.

### Target adherence

Across all restaurants and subcategories, 61% of menu items met their calorie targets (*n* = 1300/2148), 58% met their salt targets (*n* = 1348/2344), 36% met their sugar targets (*n* = 207/578), and 43% met all of their applicable targets (*n* = 1271/2951).

**By subcategory.**  Six out of the 12 subcategories had over 50% of their menu items meeting all applicable targets. Salads had the highest adherence to all applicable targets at 96% (*n* = 44/46) but were only eligible for the calorie targets (therefore having 96% adherence for calorie targets as well). Breakfast Items had the second highest adherence to all applicable targets at 66% (*n* = 125/190), while being eligible for all three target types. Excluding Other Sides and Children's Meals (which had 100% adherence to the sugar targets but each had only one eligible item), the highest sugar target adherence was seen for Breakfast Items at 74% (*n* = 56/76), which also had the highest salt target adherence at 82% (*n* = 126/154) (Fig 6).

**By restaurant.**  Nine of the 21 restaurants had over 50% of their menu items meeting all applicable targets (Fig 7).

Menu items from the Pizza restaurant group had the lowest combined adherence to all applicable targets at 32% (*n* = 310/961), to the salt targets at 49% (*n* = 409/832), and to the calorie targets at 53% (*n* = 449/846). Adherence to sugar targets was based on lower product numbers, with menu items from the Chicken restaurant group having the lowest adherence at 0%, but with only 23 eligible items. Menu items from the Pizza restaurant group were the second lowest adhering to the sugar targets at 23%, with 65 eligible items.

Menu items from the Burger restaurant group had the highest combined adherence to all applicable targets at 59% (*n* = 88/149), and to the salt targets at 80% (*n* = 92/115), while menu items from the Chicken group had the highest combined adherence to the calorie targets at 78% (*n* = 97/124). Menu items from the Burger restaurant group had the highest adherence to the sugar targets at 53%, but with only 36 eligible items, of which 32 were from McDonald's. Full results for adherence to each target type by restaurant group can be found in S19 Table.

### Sensitivity analysis excluding limited-time offer menu items

We excluded 37 limited-time offer menu items, spanning four restaurants (McDonald's, Burger King, Pret, and KFC) and eight unique subcategories, for this analysis.

The average nutrient content across all menu items did not differ from the primary analysis, except for calorie content per serving which was 1 kcal higher in the sensitivity analysis (451 kcal from 450 kcal). Overall adherence to sugar, salt, and all applicable targets did not differ from the primary analysis, but the proportion of menu items meeting calorie targets dropped by 1% (from 61% to 60%).

S20 and S21 Tables provide the average nutrient content and target adherence for the four restaurants with limited-time menu items (McDonald's, Burger King, Pret, and KFC), including and excluding limited-time offer items.

### Sensitivity analysis using lower and upper quartiles for serving size

Table 2 provides the average nutrient content across all menu items when the subcategory average, lower quartile, and upper quartile serving sizes were used to replace missing serving sizes. S22–S25 Tables provide the equivalent values for each subcategory and restaurant.

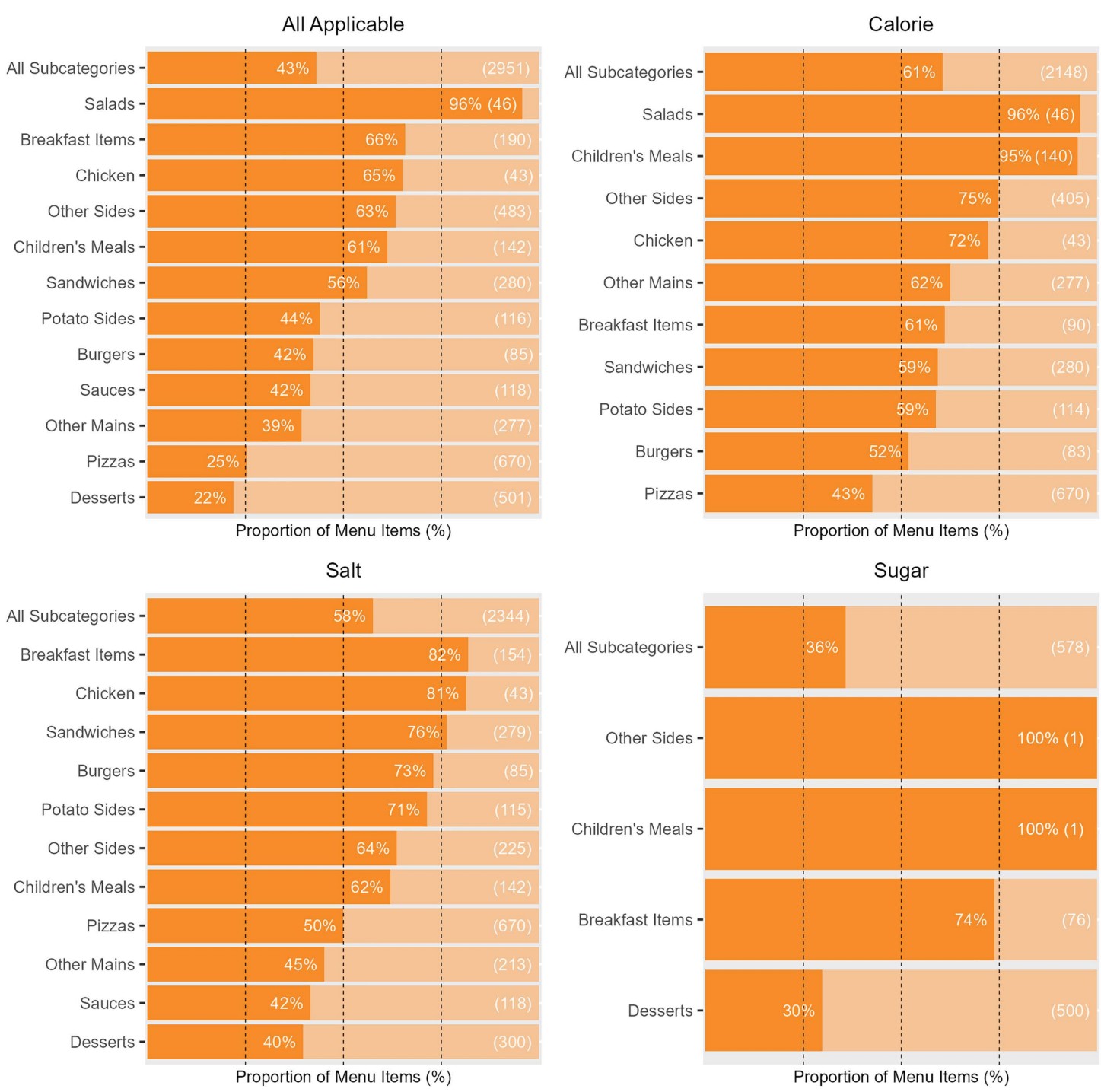

**Fig 6. The proportion of menu items that met sugar, salt, calorie, and all applicable targets, for each subcategory.** Values in brackets show the total number of menu items that were eligible for the given target, in that subcategory. Subcategories are ordered descending by the proportion of menu items meeting the respective target, with 'All Subcategories' as the top bar. The dark orange bars indicate the proportion of menu items meeting the target, and the light orange bars indicate the proportion of menu items not meeting the target.

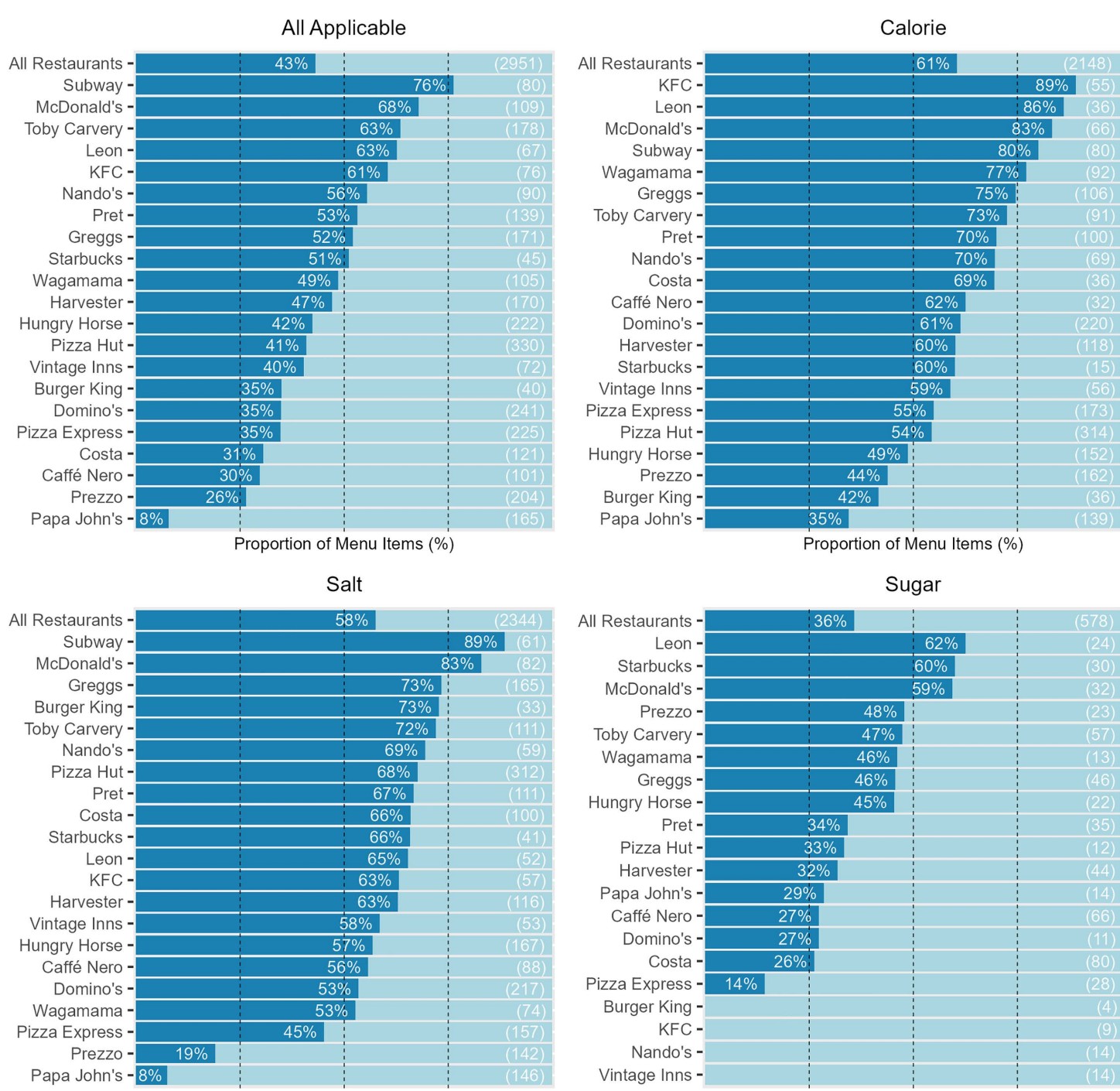

**Fig 7. The proportion of menu items that met sugar, salt, calorie, and all applicable targets, for each restaurant.** Values in brackets show the total number of menu items that were eligible for the given target, in that restaurant. Restaurants are ordered descending by the proportion of menu items meeting the respective target, with 'All Restaurants' as the top bar. The dark blue bars indicate the proportion of menu items meeting the target, and the light blue bars indicate the proportion of menu items not meeting the target.

**Table 2. Average nutrient content across all menu items when the subcategory average, lower quartile, and upper quartile serving sizes were used to replace missing serving sizes.**

| Nutrient content | Using average | Using lower quartile | Using upper quartile |
|---|---|---|---|
| kcal per 100 g | 277 | 390 | 247 |
| Salt per 100 g | 1.10 | 1.55 | 0.99 |
| Sugar per 100 g | 9.47 | 12.54 | 8.52 |
| kcal per serving | 450 | 448 | 453 |
| Salt per serving | 1.97 | 1.95 | 1.99 |
| Sugar per serving | 10.85 | 10.77 | 10.92 |

Table 3 provides the proportion of menu items meeting sugar, salt, calorie, and all applicable targets across all menu items, when the subcategory average, lower quartile, and upper quartile serving sizes were used to replace missing serving size. S26 and S27 Tables provide the equivalent values for each subcategory and restaurant.

## Discussion

This study shows that only 43% of menu items from the highest-grossing UK restaurant chains had met all of their reduction targets at the start of 2024, indicating low adherence with the reduction programmes from the OOH sector. The majority of menu items met the calorie (61%) and salt (58%) reduction targets, however, only 36% of menu items met sugar targets. Heterogeneity in adherence was observed across food categories, with Desserts having the lowest proportion of menu items meeting all applicable targets at 22%, and Salads the highest at 96%. Heterogeneity in target adherence was also observed across restaurants, with Papa John's having the lowest proportion of menu items that met all their applicable targets at 8%, and Subway the highest at 76%. A similar number of restaurants had over 50% of their menu items meeting the calorie and salt targets (17/21 and 18/21, respectively), but salt target adherence varied much more widely from 8%–89%, compared to 35%–89% for calories. Subcategories were not always consistent in their performance across the target types, for example, Children's Meals had 95% of menu items meeting calorie targets, but 62% meeting salt targets.

To our knowledge, only two studies have looked at company-level performance against the reduction targets, both focussing on the sugar reduction programme only, with one looking at manufacturers of grocery foods and the other at OOH. Bandy and colleagues [20] found that in 2018, just under half (24/50) of the best-selling manufacturers across five food categories (biscuits and cereal bars, breakfast cereals, chocolate confectionery, sugar confectionery, and yoghurts) met the intermediary 2018 sugar targets, and four companies had already met the 2020 targets. Our study found no companies had 100% of menu items meeting their sugar targets, potentially indicating greater target adherence from the retailer grocery food sector compared to OOH. However, our study included a wider range of food categories, used

**Table 3. Target adherence across all menu items when the subcategory average, lower quartile, and upper quartile serving sizes were used to replace missing serving sizes.**

| Target | Using average | Using lower quartile | Using upper quartile |
|---|---|---|---|
| Calorie | 61% | 61% | 60% |
| Salt | 58% | 56% | 59% |
| Sugar | 36% | 31% | 40% |
| All Applicable | 43% | 43% | 43% |

nutritional data from 2024 (six years later), and Bandy and colleagues used sales-weighted averages for sugar content (by brand sales volume), which we did not have the relevant data to replicate. Pepper and colleagues [21] found that 4/48 OOH companies met the 20% reduction target for Desserts in 2020, which given that Bandy and colleagues found four manufacturers had already met the 20% reduction target by 2018, this could provide another indication that the grocery sector is more engaged with the programme than OOH. Pepper and colleagues observed large variation in sugar and calorie content between companies with different styles of food, but also between similar chains, which corroborates with findings from our study across all three target nutrients (sugar, salt, and calories) and beyond just Desserts. Together, these findings suggest that while adherence to the targets may be low overall, there are examples of OOH companies that are performing well, and performance is not constrained by the type of cuisine being offered.

The most recent sugar reduction progress report found no OOH companies met the 20% reduction target by 2020 [12], and while this is consistent with our findings, it was based on data from only five companies, and four of which were sandwich/café restaurants (Costa, Pret, Greggs, and Starbucks). The most recent salt reduction progress report had insufficient data to present company-level performance for the OOH sector, although it reported that overall, 74% of OOH products met their salt target [38]. Our study found only 58% of menu items met their salt target, potentially due to the progress report using the 'maximum' target values while we used the 'average' target values, which were less lenient (e.g., Dips had a 'maximum' target of 0.9g per 100g but an 'average' target of 0.75g). There is no relevant comparison between our findings and those in the calorie reduction progress report, as this report did not include any company-level analyses [13].

Reduction target programmes in other countries have been found to be effective in reducing the sugar, salt, calorie, and fat content of food products. One systematic review found that of 26 studies evaluating government-set reduction targets across 15 different countries, 22 found improvements in nutritional quality [39]. However, the vast majority of these studies only focussed on salt, and reported the percentage change in salt content over time rather than adherence to the targets specifically, so our findings may not be directly comparable. A study assessing OOH foods in the US found significantly fewer fast-food meals met the American Heart Association's calorie guidelines in 2015, 2016, and 2017 compared to 2008, with no changes observed for saturated fat or sodium [40], suggesting that our findings are indicative of a wider global trend for poor nutritional quality in the OOH food sector. The restaurants included in our study are owned by multinational companies operating on a global scale, therefore our findings can provide insight into the nutritional quality of OOH foods beyond a UK context.

We evaluated all three reduction programmes within a single study, allowing us to compare adherence across the targets, both by individual company and overall. By providing an overview of restaurants' whole menus, we were able to demonstrate that menu items offered by restaurants with similar menu portfolios displayed variable adherence to the reduction targets. Therefore, companies should not have to change the types of foods they offer in order to improve the nutritional quality of their menus, making the shift towards a healthier OOH sector a more achievable goal for industry.

We were not able to account for heterogeneity in item-level sales due to the lack of accessible sales data from the OOH sector. It is possible that healthier menu items (items that did meet their sugar, salt, and calorie targets) are responsible for a smaller proportion of sales, thus making little difference to diet-related health outcomes. The technical guidance for the reduction targets highlights that applying the targets based on sales-weighted averages is the gold standard approach, however, this is not always possible, particularly for the OOH sector (noted in the Government's salt reduction progress report [38]). Greater transparency from companies in regards to the proportion of their sales that come from healthy and less healthy foods would permit a more holistic analysis, and could be encouraged by governments mandating the reporting of this information.

The time and resource constraints of manually collecting data from individual websites limited our sample size of menu items, and meant that the completeness and accuracy of the data was largely dependent on restaurants' own reporting. For example, we found instances of Toby Carvery underreporting kcal (detailed further in S2 Table), which we left as

reported as we did not have the capacity to laboratory test all menu items to verify the information. Only five restaurants reported serving size for all of their menu items, therefore we often had to use subcategory average serving sizes to calculate per serving or per 100 g information where missing, which may have differed to the true value given the variation we observed in nutritional content within subcategories. Not all menu items would have been included for each company, for example, we evaluated Burger King's online menu rather than their more extensive 'Nutrition Explorer', due to the less complete nutritional information in the latter, and Subway now (as of 2025) have 'Cookies and Sweet Treats' on their online menu which was not published during our data collection. These limitations reflect the lack of standardisation in reporting nutritional information across the OOH sector, highlighting another gap that government regulation could address.

We only assessed adherence at one time point, therefore we cannot determine which restaurants or subcategories have shown the most or least improvement in nutritional quality over time, which could provide further insight into which areas of the sector need more stringent monitoring.

Our data collection took place predominantly in February/March 2024, so it is possible that we would have seen better adherence to the salt targets if we had collected data later in 2024 (as the targets had to be met in 2024, with no specification of which day or month), and better adherence to the calorie targets if we had collected data later in 2025 (as the targets had to be met in 2025). However, with the lack of more recent governmental progress reports to monitor target adherence, our study provides a useful benchmark for how restaurants were performing against the targets in early 2024, which future monitoring of the targets could compare against.

The NHS 10 Year Health Plan for England [22] outlines plans for the introduction of mandatory reporting of healthy sales from large companies, with a further proposal to use this reporting to inform mandatory targets for healthy sales. Our study highlights that there is low adherence in the OOH sector with current voluntary regulation, and that monitoring of target adherence is largely limited by the lack of available sales data, which the two policies proposed in the NHS 10 Year Plan would directly address.

This study highlights the importance of mandatory reporting and targets, demonstrating overall low adherence to the targets in these voluntary programmes, with research from other countries also evidencing the increased effectiveness of mandatory (e.g., maximum limits or required declaration of salt content) versus voluntary (e.g., suggested limit on salt content) nutrition policies in inciting reformulation [39]. With no regular publication of progress reports to monitor companies' adherence to the targets, there are minimal incentives for the companies to work towards them. More regular and granular reporting of adherence to the targets by individual restaurants in the OOH sector, alongside the introduction of mandatory reporting for healthy sales for example, might lead to greater public scrutiny and thus greater adherence with the targets. While the voluntary nature of the targets may be a contributor to the low target adherence, our study was able to demonstrate that restaurants with varying cuisines are able to meet the targets, highlighting their attainability across the OOH sector.

The overall lower adherence to the sugar targets compared to the calorie and salt targets could be a result of the deadline for the sugar targets being in 2020, versus 2024/5 for the calorie and salt targets. It is possible that adherence to the sugar targets has dropped since the deadline was reached, as there has been no subsequent revisions to the sugar targets to encourage food companies to maintain their adherence to the programme. Alternatively, the sugar reduction targets were set per 100 g, while the calorie and salt targets were primarily per serving (all calorie targets per serving, OOH-specific salt targets per serving), meaning progress towards the calorie and salt targets could be made through reducing menu items' serving size, while the sugar targets could only be met through reformulation, which is potentially more time and resource intensive.

Unlike the retail sector, there is no mandatory reporting standard for nutritional information per 100 g or serving size for the OOH sector, resulting in inconsistent and limited data being used to monitor adherence to the targets. Mandating standardised nutrition reporting would improve transparency in the OOH sector and make tracking compliance to the targets easier and

more accurate, potentially inciting greater compliance from companies. Modelling work by Shangguan and colleagues [7] found the drop from 100% to 50% industry compliance with US National Salt and Sugar Reduction targets approximately halved the averted cardiovascular disease events, QALYs gained, and net savings for healthcare observed over a lifetime, highlighting the importance of meeting the targets in full in order to bring about substantial benefits to public health.

Changes could be made to the targets themselves to encourage greater adherence. All menu items had to be manually categorised into the relevant target categories to compare their nutrient content to the target value, which was time and resource-intensive, and open to subjectivity and error. Using comparable categories across targets, or setting targets based on a single holistic measure of nutritional quality rather than individual nutrients (e.g., using the UK Nutrient Profiling Model), could remove some of the burden of self-monitoring placed onto companies, permitting greater progress.

The new policy proposals in the NHS 10 Year Plan suggest there is an appetite from the UK Government to impose stricter regulation on the food sector, highlighting the importance of research to inform careful and realistic target setting, particularly within under-regulated areas of the food sector such as OOH.

In conclusion, our findings suggest there has been low adherence against the UK Government's reduction targets from the OOH sector, which could suggest that mandatory regulations may be a more effective approach to improving the nutritional quality of OOH food. While we found menu items from certain restaurant types to perform worse against the targets than others, menu items from restaurants with similar portfolios were also found to vary in target adherence, suggesting that companies should not have to change the focus of their menus in order to meet the targets, making them more attainable. This study highlights the need for standardised reporting of nutritional and serving size information from the OOH sector, alongside accessible sales data, to aid monitoring companies' performance against the targets, and in turn incite greater adherence from industry with the reduction programmes.

## Supporting information

**S1 File. Study protocol.**
(PDF)

**S1 Table. STROBE checklist.** The Strengthening the Reporting of Observational Studies in Epidemiology (STROBE) Statement: Guidelines for Reporting Observational Studies *von Elm E, Altman DG, Egger M, Pocock SJ, Gøtzsche PC, et al. (2007) The Strengthening the Reporting of Observational Studies in Epidemiology (STROBE) Statement: Guidelines for Reporting Observational Studies. PLOS Medicine 4(10): e296.* https://doi.org/10.1371/journal.pmed.0040296.
(PDF)

**S2 Table. Overview of data collection approach and completeness of collected data, for each restaurant.** Restaurants in descending order by number of menu items.
(PDF)

**S3 Table. Categorisation criteria for the 12 broad food subcategories.**
(PDF)

**S4 Table. The range of calorie, salt (per 100 g and per serving), and sugar target values set for menu items within each subcategory.**
(PDF)

**S5 Table. The number of products per subcategory where the subcategory mean serving size had to be used to calculate either per 100 g or per serving nutrient information.** Subcategories are listed in descending order by the proportion of products belonging to that category where the subcategory mean serving size had to be used.
(PDF)

**S6 Table. The number of products per restaurant where the subcategory mean serving size had to be used to calculate either per 100 g or per serving nutrient information.** Restaurants are listed in descending order by the proportion of products belonging to that category where the subcategory mean serving size had to be used.
(PDF)

**S7 Table. The mean, median, and standard deviation, for kcal per 100 g, per recommended serving, and per subcategory average serving, across all menu items in each subcategory.** In descending order by Mean kcal per 100 g.
(PDF)

**S8 Table. The mean, median, and standard deviation, for Salt per 100 g, per recommended serving, and per subcategory average serving, across all menu items in each subcategory.** In descending order by Mean Salt per 100 g.
(PDF)

**S9 Table. The mean, median, and standard deviation, for Sugar per 100 g, per recommended serving, and per subcategory average serving, across all menu items in each subcategory.** In descending order by Mean Sugar per 100 g.
(PDF)

**S10 Table. The mean, median, and standard deviation, for Fat per 100 g, per recommended serving, and per subcategory average serving, across all menu items in each subcategory.** In descending order by Mean Fat per 100 g.
(PDF)

**S11 Table. The mean, median, and standard deviation, for kcal per 100 g, per recommended serving, and per subcategory average serving, across all menu items in each restaurant.** In descending order by Mean kcal per 100 g.
(PDF)

**S12 Table. The mean, median, and standard deviation, for Salt per 100 g, per recommended serving, and per subcategory average serving, across all menu items in each restaurant.** In descending order by Mean Salt per 100 g.
(PDF)

**S13 Table. The mean, median, and standard deviation, for Sugar per 100 g, per recommended serving, and per subcategory average serving, across all menu items in each restaurant.** In descending order by Mean Sugar per 100 g.
(PDF)

**S14 Table. The mean, median, and standard deviation, for Fat per 100 g, per recommended serving, and per subcategory average serving, across all menu items in each restaurant.** In descending order by Mean Fat per 100 g.
(PDF)

**S15 Table. The mean, median, and standard deviation, for kcal per 100 g, per recommended serving, and per subcategory average serving, across all menu items in each restaurant group.** In descending order by Mean kcal per 100 g.
(PDF)

**S16 Table. The mean, median, and standard deviation, for Salt per 100 g, per recommended serving, and per subcategory average serving, across all menu items in each restaurant group.** In descending order by Mean Salt per 100 g.
(PDF)

**S17 Table. The mean, median, and standard deviation, for Sugar per 100 g, per recommended serving, and per subcategory average serving, across all menu items in each restaurant group.** In descending order by Mean Sugar per 100 g.
(PDF)

**S18 Table. The mean, median, and standard deviation, for Fat per 100 g, per recommended serving, and per subcategory average serving, across all menu items in each restaurant group.** In descending order by Mean Fat per 100 g.
(PDF)

**S19 Table. The proportion of menu items meeting sugar, salt, calorie, and all applicable targets, for each restaurant group.** In descending order by proportion of menu items meeting all applicable targets.
(PDF)

**S20 Table. Mean nutrient content per 100 g and per serving for restaurants with limited time menu items, including (as per primary analysis) and excluding the limited time offer items.**
(PDF)

**S21 Table. The proportion of menu items meeting sugar, salt, calorie, and all applicable targets, for restaurants with limited time menu items, including and excluding limited time offer items.**
(PDF)

**S22 Table. Mean nutrient content per 100 g for each subcategory when the subcategory average (as per the primary analysis), lower quartile, and upper quartile, were used to replace missing serving size.** Subcategories are listed in descending order by mean kcal per 100 g.
(PDF)

**S23 Table. Mean nutrient content per serving for each subcategory when the subcategory average (as per the primary analysis), lower quartile, and upper quartile, were used to replace missing serving size.** Subcategories are listed in descending order by mean kcal per serving.
(PDF)

**S24 Table. Mean nutrient content per 100 g for each restaurant when the subcategory average (as per the primary analysis), lower quartile, and upper quartile, were used to replace missing serving size.** Restaurants are listed in descending order by mean kcal per 100 g.
(PDF)

**S25 Table. Mean nutrient content per serving for each restaurant when the subcategory average (as per the primary analysis), lower quartile, and upper quartile, were used to replace missing serving size.** Restaurants are listed in descending order by mean kcal per 100 g.
(PDF)

**S26 Table. The proportion of menu items meeting sugar, salt, calorie, and all applicable targets for each subcategory, when the subcategory average (as per the primary analysis), lower quartile, and upper quartile, were used to replace missing serving size.** Subcategories are listed in descending order by proportion of menu items meeting all applicable targets when the average serving size was used to replace missing values.
(PDF)

**S27 Table. The proportion of menu items meeting sugar, salt, calorie, and all applicable targets for each restaurant, when the subcategory average (as per the primary analysis), lower quartile, and upper quartile, were used to replace missing serving size.** Restaurants are listed in descending order by proportion of menu items meeting all applicable targets when the average serving size was used to replace missing values.
(PDF)

## Acknowledgments

The views expressed are those of the author(s) and not necessarily those of the NIHR or the Department of Health and Social Care.

## Author contributions

**Conceptualization:** Rachel Pechey, Hannah Forde, Lauren Bandy.

**Data curation:** Alice O'Hagan.

**Formal analysis:** Alice O'Hagan.

**Funding acquisition:** Lauren Bandy.

**Investigation:** Alice O'Hagan.

**Methodology:** Alice O'Hagan, Rachel Pechey, Hannah Forde, Lauren Bandy.

**Project administration:** Lauren Bandy.

**Software:** Alice O'Hagan.

**Supervision:** Lauren Bandy.

**Validation:** Alice O'Hagan.

**Visualisation:** Alice O'Hagan.

**Writing – original draft:** Alice O'Hagan.

**Writing – review & editing:** Alice O'Hagan, Rachel Pechey, Hannah Forde, Lauren Bandy.

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
