## [Editor Report · Decision Letter 0]

10 Jul 2025

Dear Dr O'Hagan,

Thank you for submitting your manuscript entitled "Assessing adherence to the UK Government’s sugar, salt, and calorie reduction targets by the highest-grossing restaurants’ menus in 2024: A cross-sectional study" for consideration by PLOS Medicine.

Your manuscript has now been evaluated by the PLOS Medicine editorial staff and I am writing to let you know that we would like to send your submission out for external peer review.

For clinical studies, please upload a copy of your trial study protocol as a supporting information file. The study protocol should be the version submitted for approval to the institutional review board or ethics committee, should include any amendments to the study protocol, as well as the date of their approval by the institutional review or ethics committee. Please also detail any deviations from the study protocol in the Methods section of your manuscript. The editors will consider the protocol and study conduct prior to a final decision for external review.

Please re-submit your manuscript within two working days, i.e. by Jul 14 2025 11:59PM.

Kind regards,

Andreia Cunha, PhD

Senior Editor

PLOS Medicine

---

## [Decision Letter · Decision Letter 1]

11 Sep 2025

Dear Dr O'Hagan,

Sincere apologies for the delay in getting back to you with a decision, which was due to challenges in securing the necessary Reviewers. Many thanks for submitting your manuscript "Assessing adherence to the UK Government’s sugar, salt, and calorie reduction targets by the highest-grossing restaurants’ menus in 2024: A cross-sectional study" (PMEDICINE-D-25-02350R1) to PLOS Medicine. The paper has been reviewed by subject experts and a statistician; their comments are included below and can also be accessed here: [LINK]

As you will see, the reviewers find your work of considerable interest but have raised methodological concerns that would need to be fully resolved. After discussing the paper with the editorial team and an academic editor with relevant expertise, I'm pleased to invite you to revise the paper in response to the reviewers' comments. However, we would also ask you to please make the novelty of this analysis compared to previous published ones clearer in the manuscript. We plan to send the revised paper to some or all of the original reviewers, and we cannot provide any guarantees at this stage regarding publication.

We ask that you submit your revision by Dec 11 2025 11:59PM. However, if this deadline is not feasible, please contact me by email, and we can discuss a suitable alternative.

Don't hesitate to contact me directly with any questions (acunha@plos.org).

Best regards,

Andreia

Andreia Cunha, PhD

Senior editor

PLOS Medicine

acunha@plos.org

Comments from the academic editor:

A suggestion for a more appropriate title would be: Voluntary self-regulation of restaurant quality does not work: an evaluation of the UK restaurant nutrient guidelines.

Comments from the reviewers:

Reviewer #1: SUMMARY

The manuscript presents data on nutritional values and adherence to government targets for salt/sugar and calories for the 21 largest restaurant chains in the UK. Using mostly simple descriptive statistics, though also some formal null hypothesis significance testing, the authors show that adherence to targets is relatively low and characterised by substantial heterogeneity across restaurants and food categories.

PURPOSE OF REVIEW

This is a statistical review and focuses foremost on statistical rigour of the study, though I do add some less statistical comments.

OVERALL ASSESSMENT

The manuscript presents important data on nutritional values and adherence to government targets for better public health. The relevance of the work is obviously mostly limited to the UK only, though may be of interest to other countries. The reporting of analyses and results is of very high standard, complying admirably with Open Science and international reporting guidelines: the study protocol and all data are publicly available, the authors discuss deviations from the original statistical analysis plan and the manuscript largely complies with STROBE reporting guidelines.

However, there is a fundamental issue regarding the statistical analyses: the data are not a random sample from a wider population of restaurants and menu items; instead they are an exhaustive list of menu items from the 21 largest chain restaurants in the UK. As such, there is technically no point for inferential statistical analysis as there is no sampling variation - these are full population data; any difference is real in the sense that it is not due to sampling variation. This would suggest that a purely descriptive summary of the data would be more appropriate.

If one ignores this fundamental point, then there are more minor, but still quite relevant points about the statistical methods that would need addressing (see detailed comments below).

Further, a major weakness, acknowledged by the authors, is that results are not sales-weighted. This can subject results to bias as some more exotic menu items could have undue influence on the reported averages.

Finally, some analyses, particularly for nutritional values, were not particularly meaningful as they were across categories of food items that I don't think are appropriate.

All in all, I think this is a valuable study, presenting important data, the relevance of which are perhaps geographically constrained, but there are major revisions needed to the analysis and presentation of the data.

MAJOR COMMENTS

M1. My main statistical issue with the manuscript is perhaps a bit of a subtle point: inferential statistics are actually not technically applicable here as the data included in the study are not a random sample of restaurants and menu items but are an exhaustive list (with several acknowledged non-random exclusions) of menu items from the largest 21 chain restaurants. This means that where there is a difference in nutritional values or in adherence proportion, this is very much true and not subject to sampling variation (i.e. the sample means reported here are the actual population means for the top 21 restaurants included). As such, p-values are not technically meaningful here.

M2. Setting aside the more fundamental issue from comment M1, the logistic regression analyses described on l.271-283 and associated results on Table S20 and l.542-562 are not technically incorrect, but they are not the best analysis here and do not directly answer the research question ("Does adherence differ by restaurant / restaurant type?") that, I understood, the authors wanted to answer. Similar to the Kruskal-Wallis/Dunn test procedure for comparing levels of salt/sugar/calories, an omnibus test whether adherence to targets differed by restaurant or restaurant type would be more directly meaningful. An obvious candidate for this would be Fisher's exact test (or the chi-squared test if expected cell counts are large enough) for adherence and restaurant (resp. restaurant type). An alternative could be to stick with the logistic regression model, but then do a likelihood ratio test comparing the model with the restaurant dummy variables included to the null model with only an intercept term. Logistic regression would allow accounting for confounders, but that was not done anyway (and likely irrelevant here anyway), so there is not really any need for this approach. Currently the logistic regression procedure depends on choosing a reference restaurant (resp. restaurant type) and all the p-values are just whether each other restaurant (resp. restaurant type) differs from the (arbitrarily) chosen reference restaurant (resp. restaurant type) - I do not think that this is not quite what the authors set out to do. If the authors decide on taking onboard my advice for an omnibus test, then pairwise post-hoc comparisons could still be made with that approach, though I am less convinced how much value this would add beyond the descriptive summaries of the adherence results already given on Figures 6 and 7 and l.501-541.

M3. Sales-weighting would be key. While acknowledged by the authors as a limitation of the study, this is really key. Perhaps some simple online survey could be done to get some, perhaps imperfect, data on menu item popularity, that could be used for a sensitivity analysis? To illustrate the scale of the issue, e.g. from Figure 1 it is clear that McDonald has more breakfast items than burger items on their menu, but I am fairly certain they sell many more burgers than breakfasts...

SPECIFIC POINTS

S1. STROBE compliance / Table S1:

- Item 3: The objectives of the study should be more clearly stated as is currently the case. Given the focus in results on not just adherence, but also nutritional values directly, the message of the paper was not as clear as it could have been.

- Item 9: Bias needs to be more discussed as there are a number of design and analysis choices that directly bear on this, such as the inclusion of the largest 21 restaurants rather than a perhaps a more systematic approach resulting in a larger or smaller number meeting specific eligibility criteria, the lack of sales-weighting which could bias results to more exotic menu items etc.

- Item 12 d: This should probably not have been marked as 'N/A' as essentially there is no random sampling here, but an exhaustive inclusion of all menu items from the top 21 largest chain restaurants; see comment M1. So this could have been acknowledged in this section.

S2. The selection of the top 21 largest restaurant chains is a bit puzzling to me. Either a random sample from a reasonably exhaustive list of restaurants or a more systematic approach including all restaurants meeting a pre-specified set of eligibility criteria would seem a better approach to me. The reasons for including only the top 21 largest restaurants is not very clearly explained in the manuscript other than that 20 would have been a nice round number (the reason why 1 more restaurant was included in the end is explained very clearly though).

S3. The imputation strategy described on l.236-237 is in general sound, but as a single-value imputation it will introduce some bias and also underestimate uncertainty of the unknown serving size. A slightly improved version would be to fit a distribution to the sub-category serving sizes, then impute randomly from that distribution multiple times for each missing serving size, analyse the data for each imputed version of the dataset, then pool results using Rubin's rules. I acknowledge that this may be overkill here, however - unclear how much missing data there were and how wide the distributions of serving sizes per sub-category were.

S4. I am slightly sceptical of the value of the comparisons between restaurants and sub-categories described on l.241-261 and results presented on l.411-429. I would expect these to differ given that different restaurants will focus on different styles of food (some healthier than others) and clearly you would expect food sub-categories to differ anyway. The pairwise, post-hoc tests are even less meaningful in my opinion, especially as the choice to include the largest 21 restaurants is a bit arbitrary (see comment S2). Similarly, for the results on l.358-360, I do not think any of these means across sub-categories are meaningful - this is almost literally mixing apples and bananas (or sauces and chicken if you will). For example, the sauce mean values per serving are of course going to be much lower as a sauce serving is just a lot less than a burger serving. Likewise, e.g. desserts will of course be more sugary than mains and sauces, meant to give flavour, will of course be high on salt. So to some extent these comparisons are a bit trivial. I would focus on just a descriptive analysis of the nutrient values by restaurant and sub-category. This would also, I think keep a clearer focus on adherence to the government target values which is the main objective of the study. The Discussion section is firmly only on adherence results, and does not discuss the results from the statistical comparisons of nutritional values, which also suggests that there is no need for these analyses.

S5. Another key limitation, in my view, is the reliance on restaurant-reported data. This has, however, been acknowledged as a limitation by the authors on l.606-611, so I think that is fine.

S6. Language, particularly in the introduction / background section and in parts of the discussion, is a bit confusing as the authors repeatedly mention REDUCTION / CHANGE in nutritional values. But of course with a single cross-sectional timepoint, it is not possible to look at changes, but only overall levels and adherence to current government targets. This becomes quite clear in Methods and Results, but I think the authors may consider revising the introductory and discussion text. The reduction / change is the aim of the government targets, but what the study assesses is not reduction / change, but current compliance.

MINOR CLARIFICATIONS AND TYPOS

CT1. L.195: This sentence should start with "Second, [...]", given the prior use of "first, [...]" on l.188.

CT2. L.201-203: Unclear to me what the authors mean here.

CT3. L.208-212: This is a bit unclear as it seems to mix up both restaurant and item categorisations?

CT4. L.215: RStudio is just an interface to the underlying statistical computational environment R. The version number stated is an R version number not an R Studio version number. R should be cited properly as per R guidance, as should all of the packages that are listed type 'citation()' at the console for how to cite R and 'citation("dplyr")', for example, for how to cite a package (package 'dplyr' in the example).

CT5. L.256-261: This is generally statistically sound (with the caveat of my general comment M1), but can the authors clarify that Dunn's tests were only done when the overall KW test was statistically significant?

CT6. L.292: While both singular and plural are regularly used, data is more commonly (and more correctly in my opinion) used in the plural form.

CT7. L. 291-292: Please note that if sample size is large enough, there would not really be a problem with ANOVA even if the data are not normally distributed (ANOVA requires the sample means not the data to be normally distributed and due to the Central Limit Theorem, sample means will be approximately normally distributed regardless of the individual data distributions as long as the sample size is large enough; how large depends on the extent of the non-normality of the data...). I do not expect the authors to change anything in response to this comment, but just wanted to clarify that as currently stated, the reason for not doing ANOVA is really only a reason if sample sizes per level of the grouping variable are small/moderate and the non-normality is severe.

CT8. l.330: Unclear what NPM is.

CT9. Figure 1: Maybe a software restriction, but I don't think there is a need in the Desserts category to have both a total and a 'Desserts' row as they are the same (and also the heat map colour coding is irrelevant here as there is a single sub-category).

CT10. For the percentages given for the adherence results, it would be helpful if frequencies are stated as well, i.e. "x% (n/N)".

CT11. Figures 6 and 7 are very nice. In addition to the total items / restaurant stated in brackets at the end of each row, I would also add the actual count near the % figure. I would also move the note on l.520-522 of the main text into the captions for each of these two figures - that would be much clearer.

I hope the authors will find these comments helpful and that they ultimately help to improve the manuscript.

Reviewer #2: This is an important study that adds to the growing literature demonstrating that voluntary measures to improve the healthfulness of the food supply are insufficient, including in the out of home (i.e., restaurant) sector. It is well written and rigorous with important policy implications. More clarity is needed upfront about the timing of when the targets were supposed to be voluntarily adhered to, and how this relates to when the data were collected. Specific comments are below:

ABSTRACT

Line 38: It would be helpful to know in what year the UK gov set these targets for restaurants, or the year by which restaurants were supposed to meet them. I see this info is partially in the discussion but should at least be in the introduction to provide important context (and, if possible, fit into the abstract). Also, if the salt targets are not supposed to be met until 2025 ("2024/5" in line 687), and these data were collected in 2024, that is important for the reader to know upfront. It should also be acknowledged as a major limitation.

Line 41: "Progressing" implies this study will look at change, but it is cross-sectional.

Line 68 (and 578 and elsewhere): "Engagement" is not quite the right word. That is not what this study is measuring. There may have been a large increase in adherence or none. Change in adherence, which this study did not measure, would be an appropriate proxy for engagement. But because this study did not measure change, a different word is recommended. One possible suggestion is "adherence to targets."

INTRODUCTION

Line 80: Are any of these mandatory?

Line 85: What was salt intake on average in 2004 when the program was established? In other words, what % reduction was the program trying to achieve?

Line 88: What is an example of one of 24 OOH category-specific targets?

Line 109: Precisely, what percentage of energy or weekly meals comes from the OOH sector?

METHODS

Line 157: Suggest "location" instead of "site" as "site" may be misconstrued to refer to website. Also, how many were excluded because of this?

Line 175: How did the authors deal with combo/value meals? For example, for fries offered by itself AND as part of a combo, were they counted as just one item or twice?

Lines 236-9: For what % of items did you need to impute serving size or weight?

Line 330: NPM? Nutrition Profile Model?

DISCUSSION

Usually, the strengths and limitations section goes just before the conclusion and after the discussion of literature.

Line 604: If the authors agree, they could recommend additional sales-weighted targets and mandatory reporting to the government

Line 676: Can the authors provide a few specific examples of mandatory OOH regulations?

Line 683: Again, "make progress" suggests this study examined change, but it did not. Please rephrase.

To be addressed in the introduction and/or discussion: How do the methods employed herein differ from those used in other studies of OOH and nutrient changes or adherence to targets? Although this is a cross-sectional study, are there existing studies from years ago that the authors could use as a rough "baseline" to discuss changes? Or are the methods too different from those employed here?

---

* Please upload any figures associated with your paper as individual TIF or EPS files with 300dpi resolution at resubmission; please read our figure guidelines for more information on our requirements: http://journals.plos.org/plosmedicine/s/figures. While revising your submission, we strongly recommend that you use PLOS's NAAS tool (https://ngplosjournals.pagemajik.ai/artanalysis) to test your figure files. NAAS can convert your figure files to the TIFF file type and meet basic requirements (such as print size, resolution), or provide you with a report on issues that do not meet our requirements and that NAAS cannot fix.

After uploading your figures to PLOS's NAAS tool - https://ngplosjournals.pagemajik.ai/artanalysis, NAAS will process the files provided and display the results in the "Uploaded Files" section of the page as the processing is complete.

If the uploaded figures meet our requirements (or NAAS is able to fix the files to meet our requirements), the figure will be marked as "fixed" above. If NAAS is unable to fix the files, a red "failed" label will appear above.

When NAAS has confirmed that the figure files meet our requirements, please download the file via the download option, and include these NAAS processed figure files when submitting your revised manuscript.

FIGURES AND TABLES

SUPPLEMENTARY MATERIAL

REFERENCES

OBSERVATIONAL STUDIES

* Abstract: Please include the study design, population and setting, number of participants, years during which the study took place (enrollment and follow up), length of follow up, and main outcome measures.

* Please ensure that the study is reported according to the STROBE (or appropriate STOBE extension) guideline (available from: https://www.equator-network.org/reporting-guidelines/strobe) and include the completed STROBE (or STROBE extension) checklist as Supporting Information. Please add the following statement, or similar, to the Methods: "This study is reported as per the Strengthening the Reporting of Observational Studies in Epidemiology (STROBE) guideline (S1 Checklist)." When completing the checklist, please use section and paragraph numbers, rather than page numbers.

* For all observational studies, in the manuscript text, please indicate: (1) the specific hypotheses you intended to test, (2) the analytical methods by which you planned to test them, (3) the analyses you actually performed, and (4) when reported analyses differ from those that were planned, transparent explanations for differences that affect the reliability of the study's results. If a reported analysis was performed based on an interesting but unanticipated pattern in the data, please be clear that the analysis was data driven.

* Please state in the Methods section whether the study had a prospective protocol or analysis plan. If a prospective analysis plan (from your funding proposal, IRB or other ethics committee submission, study protocol, or other planning document written before analyzing the data) was used in designing the study, please include the relevant document(s) with your revised manuscript as a Supporting Information file to be published alongside your study and cite it in the Methods section. A legend for this file should be included at the end of your manuscript. If no such document exists, please make sure that the Methods section transparently describes when analyses were planned, and when/why any data-driven changes to analyses took place. Changes in the analysis, including those made in response to peer review comments, should be identified as such in the Methods section of the paper, with rationale.

---

## [Decision Letter · Decision Letter 2]

30 Jan 2026

Dear Dr. O'Hagan,

Thank you very much for re-submitting your manuscript "Voluntary UK sugar, salt, and calorie reduction targets: A cross-sectional study assessing adherence within the highest-grossing restaurants" (PMEDICINE-D-25-02350R2) for review by PLOS Medicine.

I have discussed the paper with my colleagues and the academic editor and it was also seen again by xxx reviewers. I am pleased to say that provided the remaining editorial and production issues are dealt with we are planning to accept the paper for publication in the journal.

[LINK]

We look forward to receiving the revised manuscript by Feb 06 2026 11:59PM.

Sincerely,

Andreia Cunha, PhD

Senior Editor

PLOS Medicine

plosmedicine.org

Requests from Editors:

GENERAL EDITORIAL REQUESTS

* Please confirm that your title complies with PLOS Medicine's style. Your title must be nondeclarative and not a question. It should begin with main concept if possible. "Effect of" should be used only if causality can be inferred, i.e., for an RCT. Please place the study design ("A randomized controlled trial," "A retrospective study," "A modelling study," etc.) in the subtitle (ie, after a colon).

* Please confirm that your abstract complies with our requirements, including format (three sections: Background, Methods and Findings, and Conclusions) and providing all the information relevant to this study type https://journals.plos.org/plosmedicine/s/submission-guidelines#loc-abstract

* Please ensure that the Introduction ends with a clear description of the study question or hypothesis.

* Please ensure that all abbreviations are defined at first use throughout the text.

* Please confirm that all numbers presented in the abstract are present and identical to numbers presented in the main manuscript text.

GENERAL

* Please review your text for claims of novelty or primacy (e.g. 'for the first time') and remove this language. In addition, please check that any use of statistical terms (such as trend or significant) are supported by the data, and if not please remove them.

* Statistical reporting: Please revise throughout the manuscript, including tables and figures.

- Please report statistical information as follows to improve clarity for the reader ""22% (95% CI [13,28]; p</=)"".

- Please separate upper and lower bounds with commas instead of hyphens as the latter can be confused with reporting of negative values.

- Please repeat statistical definitions (HR, CI etc.) for each set of parentheses."

* In the abstract, please include the important dependent variables that are adjusted for in the analyses.

FUNDING STATEMENT

* The funding statement should include: specific grant numbers, initials of authors who received each award, URLs to sponsors’ websites. Also, please state whether any sponsors or funders (other than the named authors) played any role in study design, data collection and analysis, the decision to publish, or preparation of the manuscript. If they had no role in the research, include this sentence: “The funders had no role in study design, data collection and analysis, decision to publish, or preparation of the manuscript.”

* It appears that one or more study authors is affiliated with one or more of the agencies that funded the study. Thus, the statement “The funders had no role in study design, data collection and analysis, decision to publish, or preparation of the manuscript” does not apply. Please revise the Financial Disclosure accordingly, as in "[Author name] is [author's role] at [funding agency]. The funders had no other role in study design…..”

COMPETING INTERESTS STATEMENT

* All authors must declare their relevant competing interests per the PLOS policy, which can be seen here: https://journals.plos.org/plosmedicine/s/competing-interests For authors with ties to industry, please indicate whether any of the interests has a financial stake in the results of the current study.

DATA AVAILABILITY

* The Data Availability Statement (DAS) requires revision. For each data source used in your study:

ETHICS AND CONSENT

* Please provide the name(s) of the institutional review board(s) that provided ethical approval. If ethics approval and/or informed consent were not required please explain this stating the reason why in the Methods section.

FIGURES

* Please define all elements of box plots in the figure caption - center line, box limits and whiskers.

* Please provide titles and legends for all figures and tables (including those in Supporting Information files). Please define all acronyms used in each figure or table in its corresponding legend.

* Please ensure that where relevant figures include 95% CIs.

OBSERVATIONAL, COHORT, CROSS-SECTIONAL, AND CASE CONTROL STUDIES

* Please ensure that the study is reported according to the STROBE guideline, and include the completed STROBE checklist as Supporting Information. Please add the following statement, or similar, to the Methods: ""This study is reported as per the Strengthening the Reporting of Observational Studies in Epidemiology (STROBE) guideline (S1 Checklist).""

* Did your study have a prospective protocol or analysis plan? Please state this (either way) early in the Methods section.

* Your study is observational and therefore causality cannot be inferred. Please remove language that implies causality and refer to associations instead.

* For all observational studies, in the manuscript text, please indicate: (1) the specific hypotheses you intended to test, (2) the analytical methods by which you planned to test them, (3) the analyses you actually performed, and (4) when reported analyses differ from those that were planned, transparent explanations for differences that affect the reliability of the study's results. If a reported analysis was performed based on an interesting but unanticipated pattern in the data, please be clear that the analysis was data-driven.

Comments from Reviewers:

Reviewer #1: The authors have fairly comprehensively addressed my comments to the original manuscript. I have no major comments remaining.

MINOR COMMENTS / TYPOS

l. 320: This is at the discretion of the editor (if there is a journal policy regarding singular or plural use of 'data') and/or the authors (if there is not): while 'data' can be found in the literature being used both as a singular and a plural, I do think the plural is more common and more correct (both in the sense that there is more than one data point in most situations and also given that 'data' is the plural of the latin 'datum') - as such, I would recommend ammending this sentence to "the data are not a random sample" or alternatively "the dataset is not a random sample".

l. 321-324: Not sure this really needs stating so explicitly in the final version of the manuscript, but if you do keep it in, then I would suggest changing the word 'true' to 'real' as I think it is slightly more appropriate. I apologise if the choice of 'true' is due to my choice of words in my original review. Even simpler, and shorter, perhaps, you could simply state that the observed differences are not due to sampling variation.

Reviewer #2: The authors were responsive to my comments and have improved the manuscript. One minor remaining item that the authors should clarify is precisely when in 2024 the salt targets were supposed to have been met. Was it January 1? Or was it October 1? This matters for understanding that extent to which the data could be considered "post-implementation" since they were collected in February-May of 2024. I do not need to re-review the manuscript as the editor can verify that this detail was added in the final version.

[LINK]

---

## [Editor Report · Decision Letter 3]

10 Feb 2026

Dear Dr. O'Hagan,

Thank you very much for re-submitting your manuscript "Voluntary UK sugar, salt, and calorie reduction targets: A cross-sectional study assessing adherence within the highest-grossing restaurants" (PMEDICINE-D-25-02350R3) for review by PLOS Medicine.

I have now carefully assessed the revisions and I am pleased to say that there are only a few remaining editorial issues that need to be dealt with before we can accept the paper for publication in the journal.

* Title: To follow PLOS Medicine guidelines, please revise the title to: Adherence to voluntary UK sugar, salt, and calorie reduction targets in the highest-grossing restaurant chains: a cross-sectional study

* Figures – please define the meaning of colours and their intensity in the figure legend.

* Please remove all subheadings from the Discussion section.

[LINK]

We expect to receive your revised manuscript within 2 days. Please email us (plosmedicine@plos.org) if you have any questions or concerns.

We look forward to receiving the revised manuscript by Feb 12 2026 11:59PM.

Sincerely,

Andreia Cunha, PhD

Senior Editor

PLOS Medicine

plosmedicine.org

[LINK]

---

## [Editor Report · Decision Letter 4]

23 Mar 2026

Dear Dr O'Hagan,

On behalf of my colleagues and the Academic Editor, Dr Barry Popkin, I am pleased to inform you that we have agreed to publish your manuscript "Adherence to voluntary UK sugar, salt, and calorie reduction targets in the highest-grossing restaurant chains: a cross-sectional study" (PMEDICINE-D-25-02350R4) in PLOS Medicine.

PRESS

Sincerely,

Andreia Cunha, PhD

Senior Editor

PLOS Medicine